# Metabolomic analysis of the occurrence of bitter fruits on grafted oriental melon plants

Shuangshuang Zhang[1,2], Lanchun Nie[1,2,3]*, Wensheng Zhao ![ORCID][1,2,3]*, Qiang Cui[1,3], Jiahao Wang[1,3], Yaqian Duan[1,3], Chang Ge[1,2]

**1** College of Horticulture, Hebei Agricultural University, Baoding, Hebei, China, **2** Hebei Key Laboratory of Vegetable Germplasm Innovation and Utilization, Baoding, Hebei, China, **3** Collaborative Innovation Center of Vegetable Industry of Hebei Province, Baoding, Hebei, China

* 13784960296@139.com (LN); zhaowensheng@hebau.edu.cn (WZ).

**Data Availability Statement:** All relevant data are within the manuscript and its Supporting Information files.

**Funding:** The authors are thankful for the financial support from the earmarked fund for Hebei

## Abstract

Grafting has been widely applied to melon (*Cucumis melo* L.) production to alleviate obstacles of continuous cropping and control soil-borne diseases. However, grafting often leads to a decline of fruit quality. For example, sometimes bitter fruits are produced on grafted plants. However, the underlying physiological mechanism still remains unclear. This study investigated the effects of different rootstocks on the taste of fruits of the Balengcui, an oriental melon cultivar, during summer production. The results showed that all grafted plants with *Cucurbita maxima* Duch. rootstocks produced bitter fruits, while non-grafted plants and plants grafted onto muskmelon rootstocks produced no bitter fruits. Liquid chromatography–mass spectrometry and metabonomic analysis were performed to investigate the mechanism underlying the occurrence of bitter fruits. Metabolite comparisons of fruits from plants grafted onto Ribenxuesong rootstocks both with non-grafted plants and plants grafted onto muskmelon rootstocks showed that 17 metabolites including phospholipids, cucurbitacins and flavonoids, exhibited changes. The three Cucurbitacins, Cucurbitacin O, Cucurbitacin C, and Cucurbitacin S, increased dramatically. The 10 phospholipids PS(18:1(9Z)/18:2(9Z,12Z)), PS(P-18:0/15:0), PA(18:1(11Z)/18:1(11Z)), PE(16:0/18:0), PS(O-16:0/17:2(9Z,12Z)), PI(16:0/18:2(9Z,12Z)), PA(15:0/22:6(4Z,7Z,10Z,13Z,16Z,19Z)), PS(P-16:0/17:2(9Z,12Z)), PS(22:0/22:1(11Z)), and PA(17:1(9Z)/0:0)) were significantly decreased, while two PA (16:0/18:2 (9Z, 12Z) and 16:0/18:1 (11Z)), two flavonoids (pelargonidin 3-(6"-malonylglucoside)-5-glucoside and malvidin 3-rutinoside) significantly increased in fruits of plants grafted onto *Cucurbita maxima* Duch. rootstocks. These metabolites were involved in the glycerophospholipid metabolic pathway, the mevalonate pathway, and the phenylpropanoid pathway. In summary, these results showed that the bitter fruits of grafted Balengcui were caused by *Cucurbita maxima* Duch. rootstocks. Phospholipids, cucurbitacins, and flavonoids were the key contributors for the occurrence of bitter fruits in Balengcui melon after grafting onto *Cucurbita maxima* Duch. rootstocks.

Vegetables Innovation Team of Modern Agro-industry Technology Research System [HBCT2018030210] to LN. The funders had no role in study design, data collection and analysis, decision to publish, or preparation of the manuscript.

**Competing interests:** The authors have declared that no competing interests exist.

## Introduction

The melon (*Cucumis melo* L.) is an important horticultural crop [1]. In 2017, the planting area was 490,327 hectares, achieving a global yield of 17,147,817 t, according to the Food and Agriculture Organization of the United Nations (FAO) [2]. Melon production plays an important role in horticultural planting [3–5]. However, intensive production led to continuous cropping obstacles and an aggravation of soil-borne diseases [6,7]. To alleviate continuous cropping obstacles and control soil-borne diseases, grafting is being widely applied to melon production [8–10]. However, grafting sometimes leads to a decline of fruit quality. Balengcui, an oriental melon cultivar known for its crispy fruit, was commonly grafted onto *Cucurbita maxima* Duch. However, the grafted plants produced bitter fruits during summer cultivation.

Cucurbitacins were identified as the main compounds that caused the bitterness of *Cucurbitaceae* fruits [11]. Highly oxidized tetracyclic triterpenoid compounds were synthesized by the methylovalerate pathway from acetyl CoA under the catalysis of a series of enzymes [12]. Generally, cultivated *Cucurbitaceae* plants do not produce bitter fruits. However, several factors, such as low or high temperature, excessive nitrogen, grafting, plant growth regulators, and changes in the physical and chemical properties of soil may lead to a coordinated operation of a number of genes and induce the synthesis of cucurbitacins [13–15]. Kano and Goto reported that during cucumber growth, plants that received twice the amount of nitrogen fertilizer had a higher probability to produce bitter leaves and fruits than plants that received a normal amount of nitrogen fertilizer [16]. Excessive use of nitrogen fertilizer led to an imbalance of the N metabolism in cucumber plants, and HMG-CoA reductase activity increased, which resulted in the synthesis of cucurbitacin in cucumber plants. However, few reports addressed how grafting caused bitterness in fruits.

Both internal and external factors can cause a coordinated physiological response. Metabolomics as a well-established method to characterize the plant metabolism has been widely used to study the effects of environmental factors on the plant physiological metabolism [17,18] and identify the response mechanisms of plants to biotic or abiotic stresses [19]. Meng *et al*. compared the metabolite profiles of high-flavonoid mutants and common Ginkgo leaves, and identified 72 different metabolites related to the flavonoid biosynthesis [20]. Metabonomics analysis indicated that both the phenylpropanoid biosynthesis pathway and lipid metabolism pathway regulated Ginkgo flavone biosynthesis. In this study, the effect of different rootstocks on the occurrence of bitter fruits of Balengcui during summer production was investigated. Liquid chromatography–mass spectrometry (LC-MS) and metabonomic analysis were performed to investigate the mechanism underlying the occurrence of bitter fruits of grafted plants.

## Materials and methods

### Plant materials and growth conditions

To investigate the effect of different rootstocks on the occurrence of bitter fruits, Balengcui seedlings with two cotyledons were grafted onto six *Cucurbita maxima* Duch. rootstocks (Yemuyixiong, Ribenxuesong, Qingshengzhenjia, Jingxinzhen3, Feichangfuzuo, and Jingyutianzhen1) and two *Cucumis melo* L. rootstocks (Inbred line1 and Inbred line2). Detailed information about the rootstocks is listed in S1 Table. Both grafted and non-grafted seedlings with 3–4 euphylla were planted in a greenhouse on June 20, 2018. This study used a randomized block design with three replications. The area of each plot was 10 m$^2$ and the planting space was 70 × 35 cm. The temperature was 35˚C during the day and 22˚C during the night and the relative air humidity was 70–80%.

## Identification of bitter fruits

During the flowering and fruit setting periods, the fruits that flowered on the same day were marked at 10–12 nodes of the main vine. At the fruit marketable mature stage, 25 fruits with the same maturity were taken per plot. Twenty tasters were selected to sample all fruits. All tasters evaluated the fruits objectively and agreed to publish their evaluation results. They rinsed their mouth after each tasting to avoid affecting their judgment of the next sample [21]. A fruit was defined as bitter when more than 10 tasters identified it as bitter. The bitter score was evaluated with a 10-point system and the average value was used.

## LC-MS analysis

Bitter fruits with the highest bitter score from grafted plants that used Ribenxuesong as root-stock, non-bitter fruits from grafted plants with muskmelon Inbred Line 1 as rootstock, and non-bitter fruits from non-grafted plants were used for metabolomic analysis. Fruit samples (60 mg) with 20 μl internal standard solutions (L-2-chloro-phenylalanine, 0.3 mg/mL) and 1 mL methanol were broken into homogenate and ultrasonically extracted for 45 min. The extracts were centrifuged and the supernatants were taken for LC-MS analysis.

LC-MS was performed with a Waters UPLC I-class system, equipped with a binary solvent delivery manager and a sample manager, and a Waters VION IMS Q-TOF Mass Spectrometer with an electrospray ionization (Waters, Milford, CT, USA), which can operate in either positive or negative ion mode. An acquity BEH C18 column (100 mm × 2.1 mm i.d., 1.7 μm) was used as chromatographic column and a temperature of 45˚C was maintained. Mobile phases A and B were water and acetonitrile, both of which included 0.1% formic acid. The flow rate was 0.40 mL/min and the injection volume was 3.00 μl. The gradient elution program is listed in the following: 5–20% B, 0–2 min, 20–60% B, 2–8 min, 60–100% B, 8–12 min, 100% B, 2 min, 100% to 5% B, 14–14.5 min, 5% B, 1 min. The temperatures of ion source and desolvation were 120˚C and 500˚C, respectively. The desolvation gas flow rate was 900 L/h. The mass spectrum scan ranged from 50 to 1,000 m/z and the scanning time was 0.1 s with an interval of 0.02 s.

## Data analysis

Progenesis QI (Waters Corporation, Milford, CI, USA) was used to perform baseline filtration, peak identification, search characterization, integration, retention time correction, peak alignment, and normalization. The secondary fragments of the metabolite detection were matched to the standard position in the respective database to obtain the data matrix consisting of retention time, mass-to-charge ratio (or metabolite name), and peak intensity. SIMCA-14.1 (Umetrics AB, Umea, Sweden) was used to implement the principal component analysis (PCA) and the orthogonal partial least-squares discrimination analysis (OPLS-DA) multivariate statistical analyses. The different metabolites were initially screened based on a variable importance in the projection (VIP) >1. Student t-test was performed by SPSS11.5 and metabolites were assumed to be significantly different at $P < 0.05$.

## Results

### Effects of different rootstocks on the occurrence of bitter fruits

The results showed that all grafted plants with *Cucurbita maxima* Duch. rootstocks produced bitter fruits at different proportions (Table 1). The highest proportion of bitter fruits was 72% in plants grafted onto Ribenxuesong rootstocks. However, no bitter fruits were identified in all plants grafted onto muskmelon rootstocks (Inbred line 1 and 2) and in all non-grafted plants.

**Table 1. Evaluation of bitter fruits.**

| Rootstocks | Number of bitter fruits | Total number of fruits | Rate of bitter fruit (%) |
|---|---|---|---|
| Yemuyixiong | 7 | 25 | 28 |
| Ribenxuesong | 18 | 25 | 72 |
| Qingshengzhenjia | 10 | 25 | 40 |
| Jingxinzhen 3 | 12 | 25 | 48 |
| Feichangfuzuo | 10 | 25 | 40 |
| Jingyutianzhen1 | 13 | 25 | 52 |
| Muskmelon Inbred line 1 | 0 | 25 | 0 |
| Muskmelon Inbred line 2 | 0 | 25 | 0 |
| Non-grafted | 0 | 25 | 0 |

These data indicated that rootstocks are responsible for the occurrence of bitter fruits of Balengcui grafted plants during summer cultivation.

## Identification and analysis of fruits metabolites

LC-MS ion flow maps of bitter fruits from plants grafted onto Ribenxuesong as rootstock, non-bitter fruits from plants grafted onto muskmelon Inbred Line 1 as rootstock, and non-bitter fruit from non-grafted plants are shown in Fig 1. The number and intensity of metabolites in fruits of plants grafted onto Ribenxuesong rootstocks were significantly different compared with fruits of non-grafted plants and plants grafted onto muskmelon rootstocks in positive and negative ~~ion~~ mode (Fig 1A and 1B). 1194 metabolites were identified (Fig 2), 27% of which were fatty acyl metabolites, 23% were glycerol phospholipids metabolites, 12% were sterol lipid metabolites, 11% were polyketide glycosides metabolites, 6% were glycerol metabolites, 5% were prenol lipid metabolites, 4% were sphingolipids metabolites, and 2% were steroid and derivative metabolites, carboxylic acid and derivative metabolites, and organic oxide metabolites. Furthermore, 1% amino acids and 5% other compounds were identified.

PCA was performed to verify the data and the PCA score map is shown in Fig 3 ($R^2X = 0.763$, $Q^2 = 0.219$). All samples were uniformly distributed in the T2 ellipse, indicating a lack of abnormal sample points. Sample points of the same group clearly clustered together and sample points from three types of fruit can be distinguished in the overall distribution. These results indicated the existence of significant differences in metabolites among fruits from non-grafted plants and plants grafted onto Ribenxuesong or muskmelon Inbred Line 1 as rootstocks. The model parameter was $R^2X = 0.763$, which indicates a good fit and can be used for the screening and analysis of different metabolites.

To further analyze the metabolites of fruits, OPLS-DA models were established. The score maps are shown in Fig 4. The OPLS-DA model parameters of fruits from plants grafted onto Ribenxuesong rootstocks and non-grafted plants were $R^2X = 0.586$, $R^2Y = 0.955$, $Q^2 = 0.873$ (Fig 4A). The parameters of fruits from plants grafted onto Ribenxuesong rootstocks and muskmelon Inbred Line 1 rootstocks were $R^2X = 0.598$, $R^2Y = 0.972$, $Q^2 = 0.881$ (Fig 4B). The parameters of $R^2$ and $Q^2$ all exceeded 0.5. Six replicates per group were closely concentrated and sample points of different groups were well separated. Therefore, these models could be used to screen different metabolites.

## Effects of different rootstocks on fruit metabolites

According to the results of OPLS-DA with the screening condition of VIP > 1 and P < 0.05, 37 metabolites, including phospholipids, sterols, flavonoids, and terpenoids, significantly

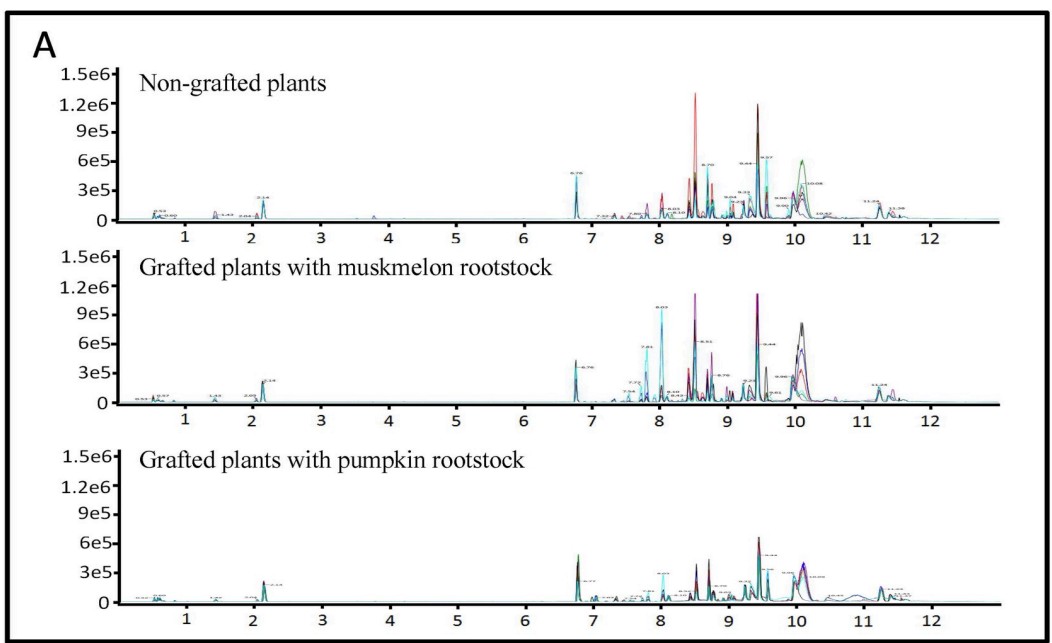

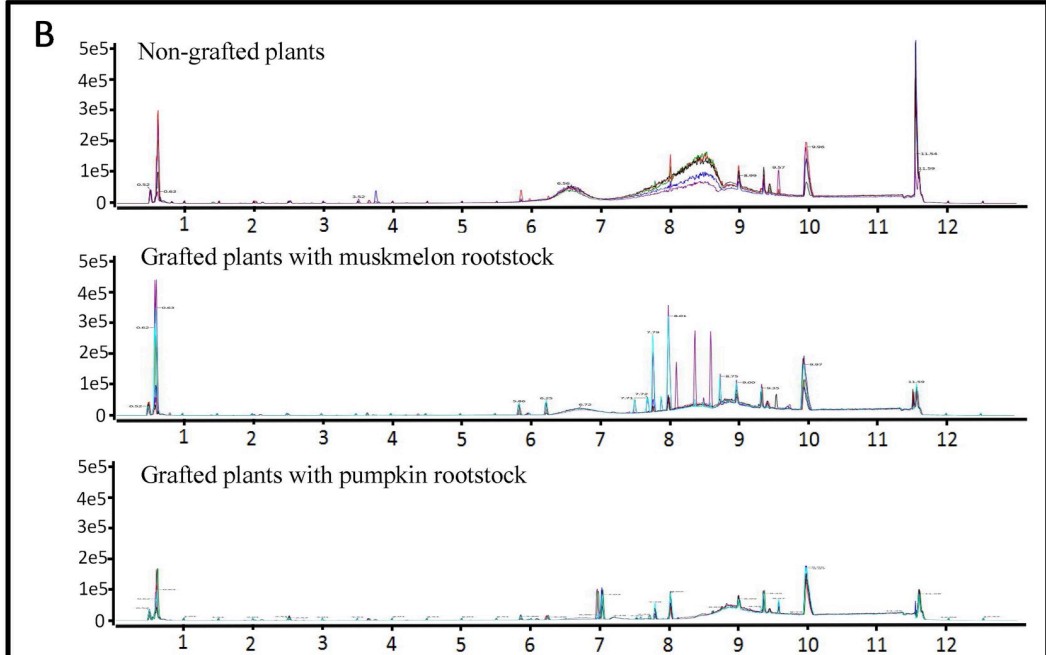

**Fig 1. Base peak intensity (BPI) chromatograms of melon fruits from non-grafted plants and plants grafted onto either muskmelon or pumpkin rootstocks.** (A) BPI chromatogram in positive mode. (B) BPI chromatogram in negative mode.

changed in bitter fruits of plants grafted onto Ribenxuesong as rootstock compared with fruits of non-grafted plants (S2 Table). Sixteen phospholipids were significantly decreased and six were significantly increased. For sterol metabolites, four were significantly increased and two were significantly decreased. Three flavonoid metabolites were significantly increased and one was significantly decreased. The terpenoids metabolites cucurbitacin O, cucurbitacin C, and cucurbitacin S increased 398.53, 203.33, and 238.21 times, respectively. Two further

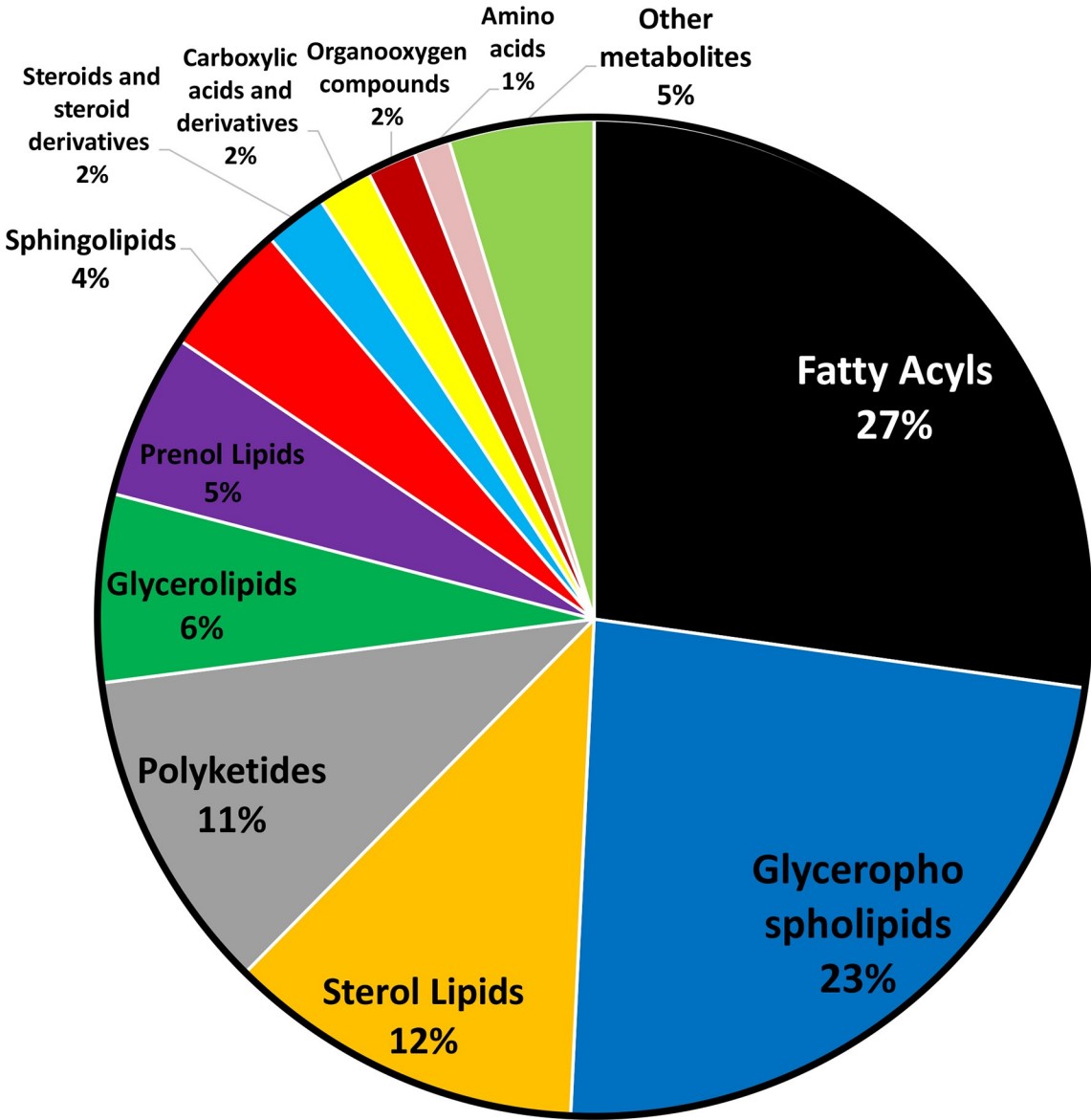

**Fig 2. Classification of identified fruit metabolites.** Different classifications of fruit metabolites are shown as different colors: Black, fatty acyls. Blue, glycerophospholipids. Orange, sterol lipids. Gray, polyketides. Green, glycerolipids. Purple, prenol lipids. Red, sphingolipids. Light blue, steroids and steroid derivatives. Yellow, carboxylicacids and derivatives. Dark red, organooxygen compounds. Pink, amino acids. Light green, other metabolites.

metabolites (spiramycin and monoisobutyl phthalic acid) were also significantly increased in fruits of plants grafted onto Ribenxuesong rootstocks.

Comparing metabolites of bitter fruits of the plants grafted onto Ribenxuesong as rootstock with fruits of plants grafted onto muskmelon Inbred Line 1 as rootstock identified 33 significantly changed metabolites including 16 phospholipids, four flavones, three sterols, three terpenoids, three organic acids, and four other compounds (S3 Table). Among these 16 phospholipid metabolites, PA (16:0/18:2 (9Z, 12Z), PA (16:0/18:1 (11Z), and PA (18:1 (9Z)/ 18:4 (6Z, 9Z, 12Z, 15Z) increased, while the other 13 glycerophospholipids decreased significantly. Among the four flavonoids, Pelargonidin 3-(6"-malonylglucoside)-5-glucoside,

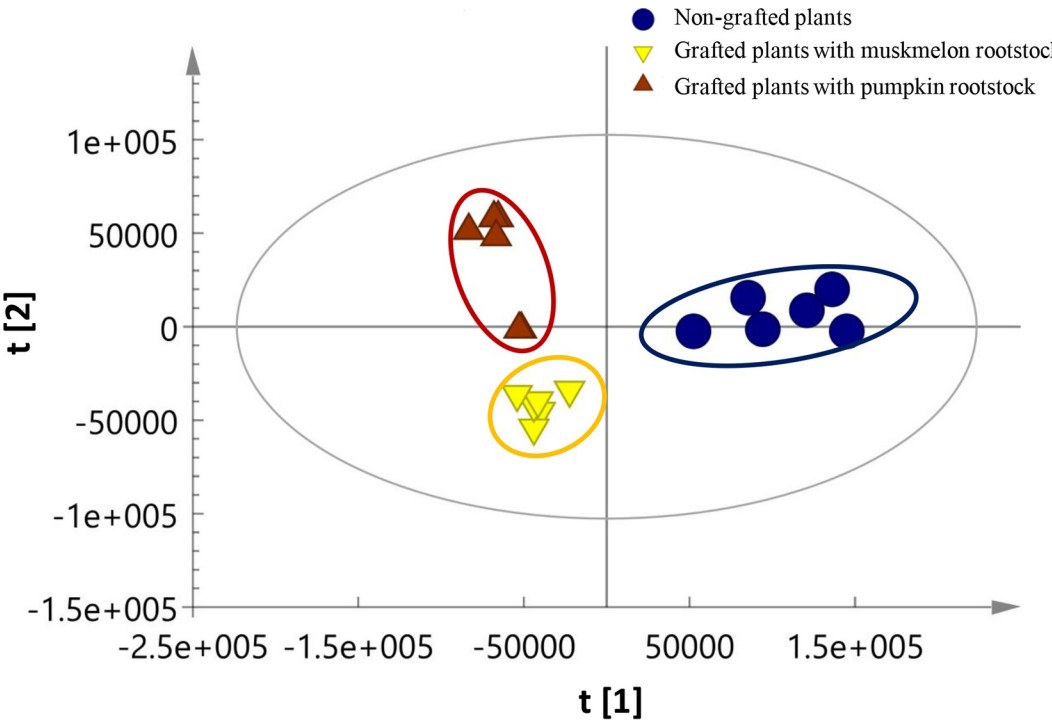

**Fig 3. PCA score plot of melon fruits from non-grafted plants and plants grafted onto muskmelon or pumpkin rootstocks.** $R^2X = 0.763$, $Q^2 = 0.219$.

Malvidin 3-rutinoside, and (+)-Myristinin A increased, while 7-Prenyloxy-3',4'-dimethoxyiso-flavone decreased. For the three sterols, hippuristanolide increased, while 1β, 3β, 5α, 6β-tetra-hydroxyandrostan-17-one, and diginatin decreased. The three terpenoids Cucurbitacin O, Cucurbitacin C, and Cucurbitacin S increased by 89.36 times, 337.81 times, and 168.58 times, respectively. 9-tetradecynoic acid decreased. Citric acid, 26:4(11Z,14Z,17Z,20Z) and glycerol esters, DG (16:0/18:3 (9Z, 12Z, 15Z)/0:0) and DG (20:5 (5Z, 8Z, 11Z, 14Z, 17Z)/0:0/20:5 (5Z, 8Z, 11Z, 14Z, 17Z)) and two other metabolites (D-Maltose and Uridine 5'-monophosphate) increased.

Overall, compared with the non-bitter fruits produced by non-grafted plants and plants grafted onto muskmelon Inbred Line 1 rootstock, 17 metabolites showed changes, including three terpenoids, two flavonoids, and 12 phospholipids in bitter fruits of plants grafted onto Ribenxuesong as rootstock (Fig 5). Phospholipids (PS(18:1(9Z)/18:2(9Z,12Z)), PS(P-18:0/ 15:0), PA(18:1(11Z)/18:1(11Z)), PE(16:0/18:0), PS(O-16:0/17:2(9Z,12Z)), PI(16:0/18:2 (9Z,12Z)), PA (15:0/22:6(4Z,7Z,10Z,13Z,16Z,19Z)), PS(P-16:0/17:2(9Z,12Z)), PS(22:0/22:1 (11Z)), and PA(17:1(9Z) /0:0)) were significantly decreased, while PA (16:0/18:2 (9Z, 12Z)) and PA (16:0/18:1 (11Z) were significantly increased (Fig 5A). The three terpenoid metabolites Cucurbitacin O, Cucurbitacin C, and Cucurbitacin S significantly increased (Fig 5B). Two fla-vonoids, pelargonidin 3-(6'-malonyl glucoside) -5-glucoside and malvidin 3-rutinoside, were significantly increased (Fig 5C). These 17 different metabolites were mapped onto KEGG and Metabo Analyst 3.0 websites. Pathway enrichment analysis was conducted by combining -log P and Impact values. The involved metabolic pathways were the glycerol phospholipid path-way, the methylovalerate pathway, and the phenylpropane pathway.

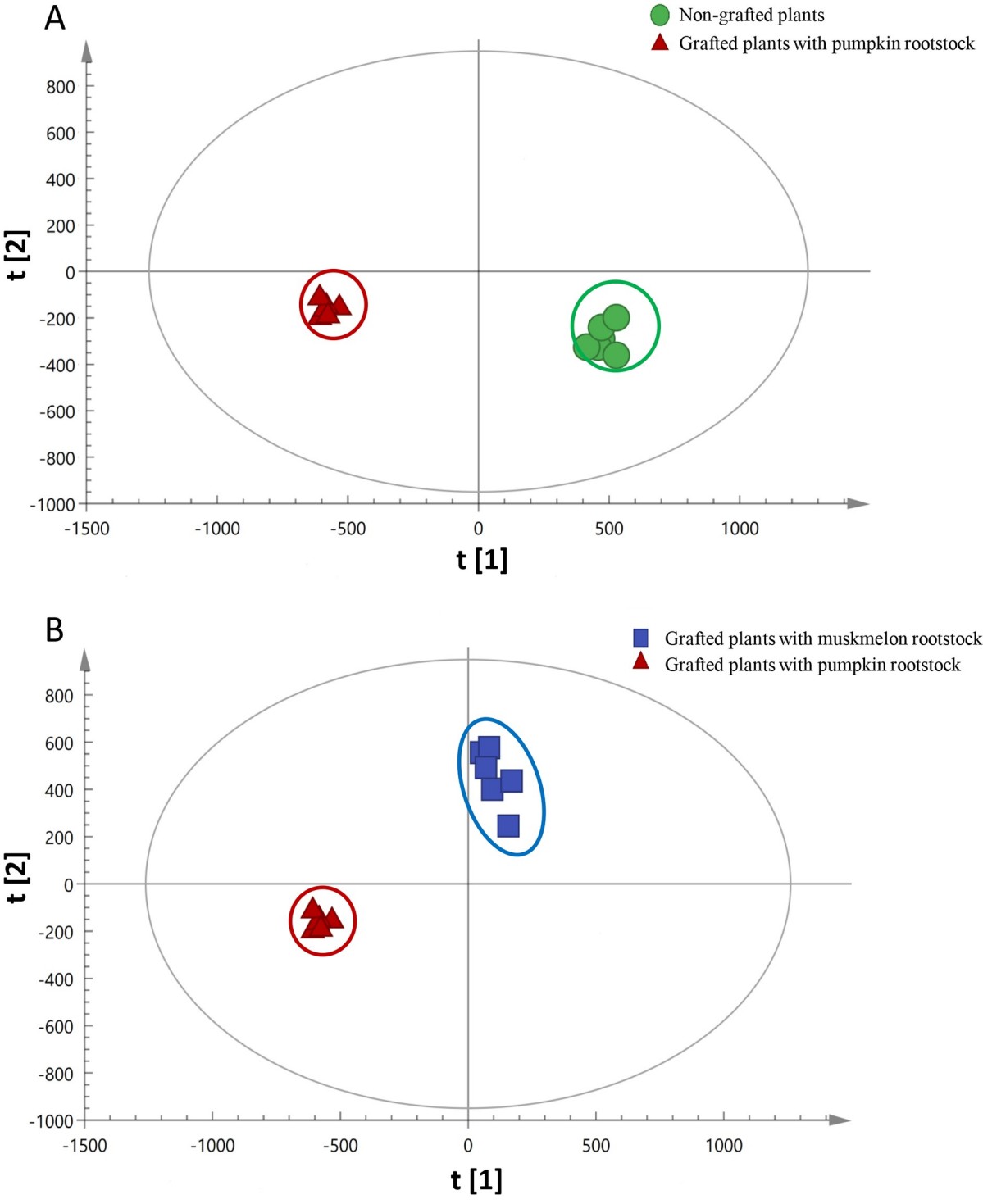

**Fig 4. OPLS-DA score plot of melon fruits from non-grafted plants and plants grafted onto muskmelon or pumpkin rootstocks.** (A) OPLS-DA score plot of melon fruits from non-grafted plants and from plants grafted onto pumpkin rootstocks. $R^2X = 0.586$, $R^2Y = 0.955$, $Q^2 = 0.873$. (B) OPLS-DA score plot of melon fruits from plants grafted onto muskmelon rootstocks and plants grafted onto pumpkin rootstocks. $R^2X = 0.598$, $R^2Y = 0.972$, $Q^2 = 0.881$.

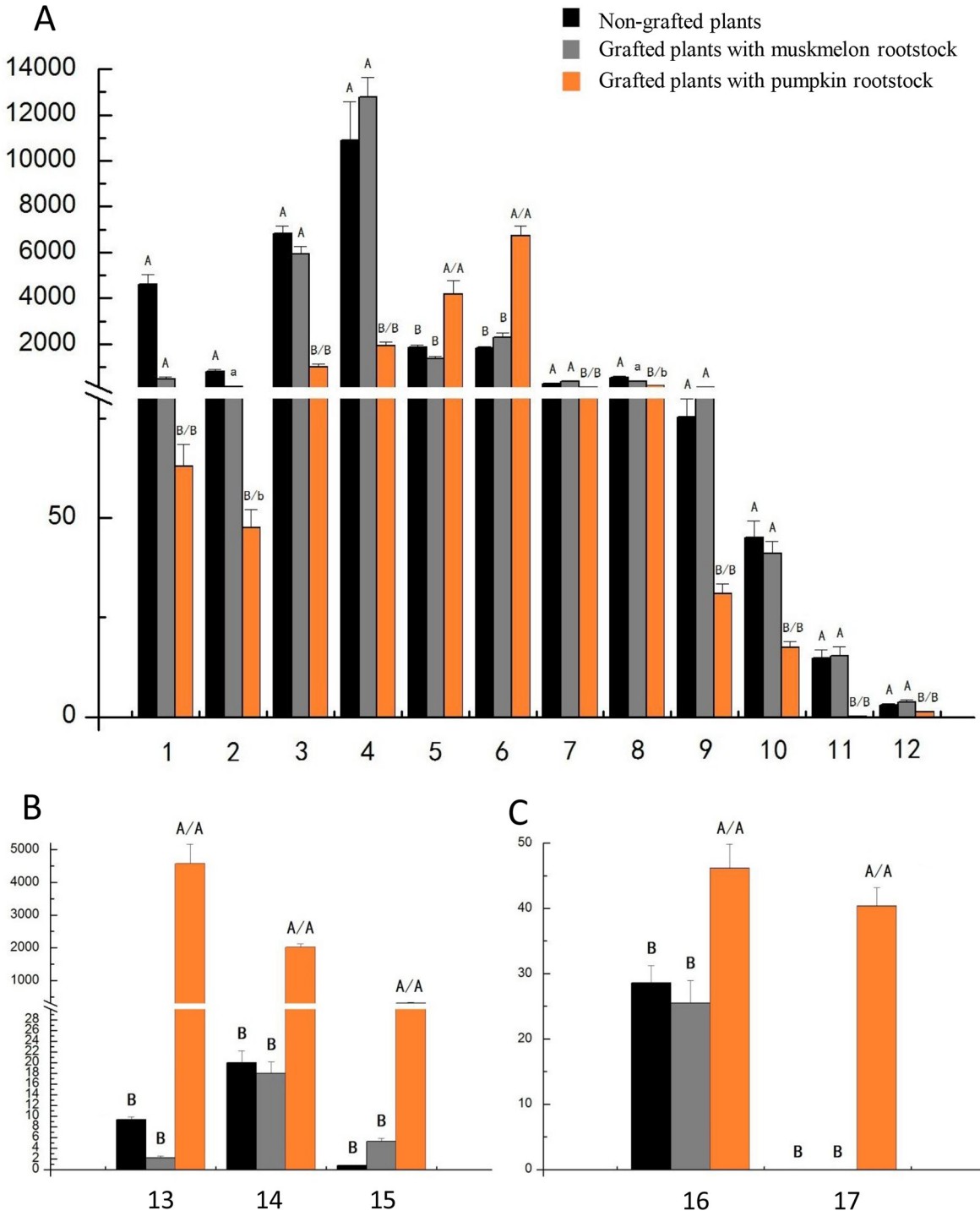

**Fig 5. Significantly changed fruit metabolites identified via comparison between plants grafted onto pumpkin rootstocks and non-grafted plants or plants grafted onto muskmelon rootstocks.** Seventeen significantly changed fruit metabolites were identified including phospholipids (A), cucurbitacin (B), and flavonoids (C). Numbers 1–12 represent PS(18:1(9Z)/18:2(9Z,12Z)), PS(P-18:0/15:0), PA(18:1 (11Z)/18:1(11Z)), PE(16:0/18:0), PA(16:0/18:1(11Z)), PA(16:0/18:2(9Z,12Z)), PS(O-16:0/17:2(9Z,12Z)), PI(16:0/18:2(9Z,12Z)), PA(15:0/ 22:6(4Z,7Z,10Z,13Z,16Z,19Z)), PS(P-16:0/17:2(9Z,12Z)), PS(22:0/22:1(11Z)), PA(17:1(9Z)/0:0), respectively. Numbers 13–15 represent Cucurbitacin C, Cucurbitacin S, and Cucurbitacin O, respectively. Numbers 16 and 17 represent pelargonidin 3-(6"-malonylglucoside)-5-glucoside and malvidin 3-rutinoside, respectively. Lowercase letters represent P < 0.05 and capital letters represent P < 0.01. The letter before "/" indicates the significance between plants grafted onto pumpkin rootstocks and non-grafted plants. The letter after "/" indicates the significance between plants grafted onto pumpkin rootstocks and muskmelon rootstocks.

## Discussion

Bitterness sometimes occurs in fruits of cucurbitaceae plants such as melon, cucumber, and gourd, which severely affected the fruit quality. Low or high temperature, malnutrition, plant growth regulators, and rootstock could all lead to bitter fruits in cucurbitaceae plants [14, 15]. In this study, all Balengcui plants grafted onto *Cucurbita maxima* Duch. produced bitter fruits, but bitter fruits were not produced on the non-grafted plants and plants grafted onto musk-melon rootstocks. These *Cucurbita maxima* Duch. rootstocks were also often used as Baleng-cui rootstocks by farmers during winter; however, these grafted plants did not produce bitter fruits. Why bitter fruits were only produced in summer remained unknown.

LC-MS was performed for metabonomic analysis to investigate the mechanism underlying the occurrence of bitter fruits. Metabolites comparisons of fruits from the plants grafted onto Ribenxuesong rootstocks both with non-grafted plants and plants grafted onto muskmelon rootstocks showed that 17 metabolites exhibited the same changes (Fig 5). Three cucurbitacins dramatically increased hundreds of times in the fruits of plants grafted onto Ribenxuesong rootstocks. Cucurbitacins, as secondary metabolites of highly oxidized tetracyclic triterpenoids, were the main compounds that caused bitterness in fruits of cucurbitaceous plants such as cucumber, pumpkin, and squash [22–24]. These metabolites were the direct cause of bitterness of fruits from plants grafted onto Ribenxuesong rootstocks.

This increase of cucurbitacins was accompanied by a significant decrease of 10 phospholipids, and increases of two PA (PA (16:0/18:1 (11Z) and PA (16:0/18:2 (9Z, 12Z)) and two flavonoids (pelargonidin 3-(6"-malonylglucoside)-5-glucoside and malvidin 3-rutinoside) in the fruits of plants grafted onto Ribenxuesong rootstocks (Fig 5A and 5C). Phospholipids are important membrane components, which are essential for the maintenance of cell stability and for the protection of plants against stress [25]. PAs are also important lipid signaling molecules that can be found in plants, which involved mediating the production and reaction of peroxides and affecting the accumulation of oxidized lipids [26–28]. Several PAs increased under abiotic stress in plants [29]. Flavonoids, as secondary metabolites of phenolic compounds, play an important role in the plant response to environmental signals. Flavonoids can protect plant tissues from damage of high-energy wavelengths (ultraviolet-A and ultraviolet-B) under high temperature and strong light stress [30, 31].

The decrease of a large number of phospholipids and the increase of two PAs in the fruits of plants grafted onto Ribenxuesong rootstocks indicated that fruits were subjected to stress (high temperature and strong light in summer), which led to the occurrence of and membrane lipid peroxidation. The flavonoid increase was also in response to this stress. Wu *et al*. reported that grafted muskmelon plants with pumpkin rootstocks exhibited lower antioxidant enzyme activity and a higher level of reactive oxygen species (ROS) [32, 33]. ROS lead to membrane lipid peroxidation, which destroyed the cell membrane structure and physiologic integrity [34, 35]. Pumpkins are biologically less resistant to high temperatures than melons. During summer cultivation, the prevailing high temperature and strong light stress may have caused an imbalance of the ROS metabolism system in fruits of plants grafted onto pumpkin rootstocks. This then resulted in membrane lipid peroxidation and phospholipid decomposition into fatty acids [36]. These fatty acids were further oxidized and produced large amounts of acetyl coenzyme A, which (as a precursor) entered the methylovalerate pathway and Cucurbitacins were synthesized [37]. However, the molecular mechanism underlying the signal transduction and gene regulation of the cucurbitacin synthesis through the methylovalerate pathway and how the rootstocks affect the fruit metabolism still requires further research.

## Conclusions

Pumpkin (*Cucurbita maxima* Duch.) rootstocks yileded bitter fruits in grafted melon plants during summer production. In bitter fruits, dramatic increases of cucurbitacins were accompanied by significant decreases of a large number of phospholipids and increases of several PAs and flavonoids. These results indicated that during summer cultivation, oxidation and decomposition of phospholipids affected the fruits of plants grafted onto pumpkin rootstocks and the methylovalerate pathway was activated to synthesize cucurbitacins. This study enables a better understanding of the mechanisms of how bitter fruits are produced on plants grafted onto specific rootstocks. Grafting onto pumpkin rootstocks for summer production should be avoided and other special rootstocks should be selected and bred.

## Supporting information

**S1 Table. Information of different rootstocks.**
(DOC)

**S2 Table. Comparison of metabolites in fruits of plants grafted onto Ribenxuesong rootstocks and non-grafted plants.**
(DOC)

**S3 Table. Comparison of metabolites in fruits of plants grafted onto Ribenxuesong and muskmelon rootstocks.**
(DOC)

## Acknowledgments

The authors are grateful to the members of the laboratory for technical assistance and encouragement.

## Author Contributions

**Conceptualization:** Shuangshuang Zhang, Lanchun Nie, Wensheng Zhao.

**Data curation:** Shuangshuang Zhang, Lanchun Nie, Wensheng Zhao, Qiang Cui, Jiahao Wang, Chang Ge.

**Formal analysis:** Shuangshuang Zhang.

**Funding acquisition:** Lanchun Nie.

**Investigation:** Shuangshuang Zhang, Lanchun Nie, Wensheng Zhao, Jiahao Wang, Yaqian Duan.

**Methodology:** Lanchun Nie.

**Project administration:** Lanchun Nie, Wensheng Zhao.

**Resources:** Lanchun Nie.

**Supervision:** Lanchun Nie.

**Validation:** Shuangshuang Zhang, Lanchun Nie, Wensheng Zhao, Qiang Cui, Yaqian Duan, Chang Ge.

**Visualization:** Shuangshuang Zhang, Lanchun Nie, Wensheng Zhao, Qiang Cui, Jiahao Wang.

**Writing – original draft:** Shuangshuang Zhang, Lanchun Nie, Wensheng Zhao.

**Writing – review & editing:** Lanchun Nie, Wensheng Zhao.

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
