## [Decision Letter · Decision Letter 0]

30 Aug 2019

[EXSCINDED]

PONE-D-19-19662

Metabolomic analysis for the occurrence of bitter fruits of grafting oriental melon plants

PLOS ONE

Dear Dr. Zhao,

Thank you for submitting your manuscript to PLOS ONE. After careful consideration, we feel that it has merit but does not fully meet PLOS ONE’s publication criteria as it currently stands. Therefore, we invite you to submit a revised version of the manuscript that addresses the points raised during the review process.

We would appreciate receiving your revised manuscript by Oct 14 2019 11:59PM. To enhance the reproducibility of your results, we recommend that if applicable you deposit your laboratory protocols in protocols.io, where a protocol can be assigned its own identifier (DOI) such that it can be cited independently in the future. For instructions see: http://journals.plos.org/plosone/s/submission-guidelines#loc-laboratory-protocols

We look forward to receiving your revised manuscript.

Kind regards,

Yuan Huang

Academic Editor

PLOS ONE

Journal Requirements:

2. We understand that no formal ethical approval was obtained for this research even though it included the participation of 20 tasters.

Please clarify if your institutional review board (IRB) waived the need to obtain formal approval and consent from participants. If participant consent was obtained, please amend your methods section to state this and specify whether the consent was informed and if verbal or written.

Reviewers' comments:

Reviewer's Responses to Questions

**Comments to the Author**

1. Is the manuscript technically sound, and do the data support the conclusions?

Reviewer #1: Yes

2. Has the statistical analysis been performed appropriately and rigorously? 

Reviewer #1: Yes

3. Have the authors made all data underlying the findings in their manuscript fully available?

Reviewer #1: Yes

4. Is the manuscript presented in an intelligible fashion and written in standard English?

Reviewer #1: No

5. Review Comments to the Author

Reviewer #1: Comments

1. The manuscript presents some interesting information that may be published after necessary revision.

2. Generally document read well but requires language editing before publication, the word grafting and grafted are used in a misappropriate way, please check. Similarly there are several other language mistake and formatting errors, please check carefully

3. Please check the following

L 1. The word “grafting” should be replaced with “grafted”

L 24-25. “For example, bitter fruits sometimes occurred on grafted plants.” Replace with “For example, sometimes bitter fruits are produced on grafted plants.”

L 164-166. Please check the word “ion”, this do not seems appropriate word for description of metabolites figure.

4. Please add a conclusion sentence at the end of abstract.

5. L 55. Following recent references may be cited

1. Mu Xiong, Xuejun Zhang, Sergey Shabala, Lana Shabala, Yanjun Chen, Chengli Xiang, Muhammad Azher Nawaz, Zhilong Bie, Haibo Wu, Hongping Yi, Mingzhu Wu, Yuan Huang. 2018. Evaluation of salt tolerance and contributing ionic mechanism in nine Hami melon landraces in Xinjiang, China. Scientia Horticulturae. 237:277-286.

https://www.sciencedirect.com/science/article/pii/S0304423818302723

2. Muhammad Azher Nawaz, Muhamamd Imtiaz, Qiusheng Kong, Cheng Fei, Waqar Ahmed, Yuan Huang and Zhilong Bie. (2016). Grafting: a technique to modify ion accumulation in horticultural crops. Frontiers in Plant Science. 7: 1457. (IF: 4.298). http://journal.frontiersin.org/article/10.3389/fpls.2016.01457/full

3. Zhilong Bie, Muhammad Azher Nawaz, Yuan Huang，Jung-Myung Lee and Giuseppe Colla. 2017. Introduction to Vegetable Grafting. In. Giuseppe Colla, Francisco Perez Alfocea, Dietmar Schwarz (Eds.). Vegetable Grafting. Principles and Practices. CABI Publishing, UK. pp. 1-21.

6. L 57. “Grafted on” should be replaced with “grafted onto”, please check and replace this throughout the manuscript.

7. Discussion section requires improvement.

8. Conclusion section just seems a summary of results, please improve and add a conclusive statement and future aspects of research in this field.

9. Please supply a high quality Figures particularly Figure 1.

6. PLOS authors have the option to publish the peer review history of their article (what does this mean?). If published, this will include your full peer review and any attached files.

Reviewer #1: Yes: Muhammad Azher Nawaz

---

## [Author Response · Author response to Decision Letter 0]

20 Sep 2019

Dear editor,

We appreciate these comments and revised the manuscript as suggested by the academic editor and reviewer. We hope that the revised manuscript has fully addressed the comments so that it is now suitable for publication in PLOS ONE. Details were listed: 

Thank you for submitting your manuscript to PLOS ONE. After careful consideration, we feel that it has merit but does not fully meet PLOS ONE’s publication criteria as it currently stands. Therefore, we invite you to submit a revised version of the manuscript that addresses the points raised during the review process.

---Response：We appreciate this comment. We have revised our manuscript and addressed each point raised during the review process.

We would appreciate receiving your revised manuscript by Oct 14 2019 11:59PM. ---Response：Thank you.

---Response：Thank you. No changes were made to our financial disclosure.

To enhance the reproducibility of your results, we recommend that if applicable you deposit your laboratory protocols in protocols.io, where a protocol can be assigned its own identifier (DOI) such that it can be cited independently in the future. For instructions see: http://journals.plos.org/plosone/s/submission-guidelines#loc-laboratory-protocols

---Response：We appreciate this comment. However, the methods used in this study were routine metabolomics analysis and there were no new approaches or insights. Therefore, we chose not deposit our laboratory protocols in protocols.io. If we develop new protocols in the future, we'd like to share it to protocols.io.

• A rebuttal letter that responds to each point raised by the academic editor and reviewer(s). This letter should be uploaded as separate file and labeled 'Response to Reviewers'.

• A marked-up copy of your manuscript that highlights changes made to the original version. This file should be uploaded as separate file and labeled 'Revised Manuscript with Track Changes'.

• An unmarked version of your revised paper without tracked changes. This file should be uploaded as separate file and labeled 'Manuscript'.

---Response：Thank you. All above-mentioned files were submitted.

---Response：Thank you. We are willing to make the peer review history publicly available.

Journal Requirements:

 ---Response：Thank you. We have proofread our manuscript carefully to meet PLOS ONE's style requirements.

2. We understand that no formal ethical approval was obtained for this research even though it included the participation of 20 tasters.

Please clarify if your institutional review board (IRB) waived the need to obtain formal approval and consent from participants. If participant consent was obtained, please amend your methods section to state this and specify whether the consent was informed and if verbal or written.

---Response：We appreciate this comment. Our institutional review board don’t need to obtain formal approval and consent from participants and all tasters have known that the evaluate results may be published. We added the sentence “All tasters evaluated the fruits objectively and agreed to publish their evaluation results.” at methods section.

Reviewers' comments:

Reviewer's Responses to Questions

Comments to the Author

1. Is the manuscript technically sound, and do the data support the conclusions?

Reviewer #1: Yes

2. Has the statistical analysis been performed appropriately and rigorously? 

Reviewer #1: Yes

3. Have the authors made all data underlying the findings in their manuscript fully available?

Reviewer #1: Yes

4. Is the manuscript presented in an intelligible fashion and written in standard English?

Reviewer #1: No

 ---Response：Thank you. We appreciate this comment and checked the whole manuscript seriously. We have also asked a colleague who is a native English speaker to carefully proofread the manuscript. We believe that all grammar problems have been corrected.

5. Review Comments to the Author

Reviewer #1: Comments

1. The manuscript presents some interesting information that may be published after necessary revision.

---Response：We appreciate this comment and have carefully revised our manuscript.

2. Generally document read well but requires language editing before publication, the word grafting and grafted are used in a misappropriate way, please check. Similarly there are several other language mistake and formatting errors, please check carefully

---Response：We appreciate this comment. In revised manuscript, the word grafting and grafted were used in an appropriate way. We have carefully checked each sentence in the manuscript and corrected all language mistake and formatting errors. For example: 

L18. “The cause of bitterness produced after grafting” was replaced with “The cause of bitterness of fruits on grafted plants”.

L198 “In order to further analyze the fruits metabolites, the orthogonal partial least-squares discrimination analysis (OPLS-DA) models were established.” was replaced with “To further analyze the metabolites of fruits, OPLS-DA models were established.”

L463. “Information of different grafted rootstocks” was replaced with “Information of different rootstocks”.

3. Please check the following

L 1. The word “grafting” should be replaced with “grafted”

L 24-25. “For example, bitter fruits sometimes occurred on grafted plants.” Replace with “For example, sometimes bitter fruits are produced on grafted plants.”

L 164-166. Please check the word “ion”, this do not seems appropriate word for description of metabolites figure.

---Response：We appreciate this comment and revised as suggested. 

L 1. The word “grafting” was replaced with “grafted”

L 24-25. “For example, bitter fruits sometimes occurred on grafted plants.” was replace with “For example, sometimes bitter fruits are produced on grafted plants.”

L 164-166. “Total ion current maps” was replaced with “Base peak intensity (BPI) chromatograms” after referring to other researches. 

4. Please add a conclusion sentence at the end of abstract.

---Response：We appreciate this comment and added the conclusion sentence “In summary, these results showed that the bitter fruits of grafted Balengcui were caused by Cucurbita maxima Duch. rootstocks. Phospholipids, cucurbitacins, and flavonoids were the key contributors for the occurrence of bitter fruits in Balengcui melon after grafting onto Cucurbita maxima Duch. rootstocks.” at the end of abstract.

5. L 55. Following recent references may be cited

1. Mu Xiong, Xuejun Zhang, Sergey Shabala, Lana Shabala, Yanjun Chen, Chengli Xiang, Muhammad Azher Nawaz, Zhilong Bie, Haibo Wu, Hongping Yi, Mingzhu Wu, Yuan Huang. 2018. Evaluation of salt tolerance and contributing ionic mechanism in nine Hami melon landraces in Xinjiang, China. Scientia Horticulturae. 237:277-286.

https://www.sciencedirect.com/science/article/pii/S0304423818302723

2. Muhammad Azher Nawaz, Muhamamd Imtiaz, Qiusheng Kong, Cheng Fei, Waqar Ahmed, Yuan Huang and Zhilong Bie. (2016). Grafting: a technique to modify ion accumulation in horticultural crops. Frontiers in Plant Science. 7: 1457. (IF: 4.298). http://journal.frontiersin.org/article/10.3389/fpls.2016.01457/full

3. Zhilong Bie, Muhammad Azher Nawaz, Yuan Huang，Jung-Myung Lee and Giuseppe Colla. 2017. Introduction to Vegetable Grafting. In. Giuseppe Colla, Francisco Perez Alfocea, Dietmar Schwarz (Eds.). Vegetable Grafting. Principles and Practices. CABI Publishing, UK. pp. 1-21.

---Response：We appreciate this comment and cited above-mentioned references as suggested.

6. L 57. “Grafted on” should be replaced with “grafted onto”, please check and replace this throughout the manuscript.

---Response：Thank you. We replaced “grafted on” with “grafted onto” in revised manuscript and checked the whole manuscript carefully as suggested.

7. Discussion section requires improvement.

---Response：We appreciate this comment. Discussion section was modified and improved in revised manuscript.

8. Conclusion section just seems a summary of results, please improve and add a conclusive statement and future aspects of research in this field.

---Response：We appreciate this comment and improved the conclusion section. Especially, We add a conclusive statement “These results indicated that during summer cultivation, oxidation and decomposition of phospholipids affected the fruits of plants grafted onto pumpkin rootstocks and the methylovalerate pathway was activated to synthesize cucurbitacins.”and future aspects of research in this field “This study enables a better understanding of the mechanisms of how bitter fruits are produced on plants grafted on specific rootstocks. Grafting onto pumpkin rootstocks for summer production should be avoided and other special rootstocks should be selected and bred.” in Conclusion section.

9. Please supply a high quality Figures particularly Figure 1.

 ---Response：We appreciate this comment and high quality Figures were supplied in revised manuscript. 

6. PLOS authors have the option to publish the peer review history of their article (what does this mean?). If published, this will include your full peer review and any attached files.

Do you want your identity to be public for this peer review? For information about this choice, including consent withdrawal, please see our Privacy Policy.

Reviewer #1: Yes: Muhammad Azher Nawaz

 ---Response：Thank you. All figure files were uploaded to PACE and no Image Problems were discovered. In revised manuscript, all figures were adjusted according to PACE Adjustments.

---

## [Decision Letter · Decision Letter 1]

25 Sep 2019

PONE-D-19-19662R1

Metabolomic analysis of the occurrence of bitter fruits on grafted oriental melon plants

PLOS ONE

Dear Dr. Zhao,

Thank you for submitting your manuscript to PLOS ONE. After careful consideration, we feel that it has merit but does not fully meet PLOS ONE’s publication criteria as it currently stands. Therefore, we invite you to submit a revised version of the manuscript that addresses the points raised during the review process.

We would appreciate receiving your revised manuscript by Nov 09 2019 11:59PM. To enhance the reproducibility of your results, we recommend that if applicable you deposit your laboratory protocols in protocols.io, where a protocol can be assigned its own identifier (DOI) such that it can be cited independently in the future. For instructions see: http://journals.plos.org/plosone/s/submission-guidelines#loc-laboratory-protocols

We look forward to receiving your revised manuscript.

Kind regards,

Yuan Huang

Academic Editor

PLOS ONE

Reviewers' comments:

Reviewer's Responses to Questions

**Comments to the Author**

1. If the authors have adequately addressed your comments raised in a previous round of review and you feel that this manuscript is now acceptable for publication, you may indicate that here to bypass the “Comments to the Author” section, enter your conflict of interest statement in the “Confidential to Editor” section, and submit your "Accept" recommendation.

Reviewer #1: All comments have been addressed

2. Is the manuscript technically sound, and do the data support the conclusions?

Reviewer #1: Yes

3. Has the statistical analysis been performed appropriately and rigorously? 

Reviewer #1: Yes

4. Have the authors made all data underlying the findings in their manuscript fully available?

Reviewer #1: Yes

5. Is the manuscript presented in an intelligible fashion and written in standard English?

Reviewer #1: Yes

6. Review Comments to the Author

Reviewer #1: Line 337-338: Please replace "grafted on specific" with "grafted onto specific"

Please revise acknowledgement section "technical assistance and stimulating discussions" may be replaced with "technical assistance and encouragement"

7. PLOS authors have the option to publish the peer review history of their article (what does this mean?). If published, this will include your full peer review and any attached files.

Reviewer #1: Yes: Muhammad Azher Nawaz

---

## [Author Response · Author response to Decision Letter 1]

25 Sep 2019

Dear editor,

We appreciate these comments and revised the manuscript as suggested. We hope that the revised manuscript has fully addressed the comments so that it is now suitable for publication in PLOS ONE. 

Details were listed: 

“Dear Dr. Zhao,

Thank you for submitting your manuscript to PLOS ONE. After careful consideration, we feel that it has merit but does not fully meet PLOS ONE’s publication criteria as it currently stands. Therefore, we invite you to submit a revised version of the manuscript that addresses the points raised during the review process.

We would appreciate receiving your revised manuscript by Nov 09 2019 11:59PM. To enhance the reproducibility of your results, we recommend that if applicable you deposit your laboratory protocols in protocols.io, where a protocol can be assigned its own identifier (DOI) such that it can be cited independently in the future. For instructions see: http://journals.plos.org/plosone/s/submission-guidelines#loc-laboratory-protocols

• A rebuttal letter that responds to each point raised by the academic editor and reviewer(s). This letter should be uploaded as separate file and labeled 'Response to Reviewers'.

• A marked-up copy of your manuscript that highlights changes made to the original version. This file should be uploaded as separate file and labeled 'Revised Manuscript with Track Changes'.

• An unmarked version of your revised paper without tracked changes. This file should be uploaded as separate file and labeled 'Manuscript'.

We look forward to receiving your revised manuscript.

Kind regards,

Yuan Huang

Academic Editor

PLOS ONE”

---Response：We appreciate these comments. We have revised our manuscript and addressed each point raised during the review process. No changes were made to our financial disclosure and we chose not deposit our laboratory protocols in protocols.io. All needed files including 'Response to Reviewers', 'Revised Manuscript with Track Changes' and 'Manuscript' were submitted and we are willing to make the peer review history publicly available if the revised manuscript is accepted.

Reviewers' comments:

Reviewer's Responses to Questions

Comments to the Author

1. If the authors have adequately addressed your comments raised in a previous round of review and you feel that this manuscript is now acceptable for publication, you may indicate that here to bypass the “Comments to the Author” section, enter your conflict of interest statement in the “Confidential to Editor” section, and submit your "Accept" recommendation.

Reviewer #1: All comments have been addressed

2. Is the manuscript technically sound, and do the data support the conclusions?

Reviewer #1: Yes

3. Has the statistical analysis been performed appropriately and rigorously? 

Reviewer #1: Yes

4. Have the authors made all data underlying the findings in their manuscript fully available?

Reviewer #1: Yes

5. Is the manuscript presented in an intelligible fashion and written in standard English?

Reviewer #1: Yes

6. Review Comments to the Author

Reviewer #1: Line 337-338: Please replace "grafted on specific" with "grafted onto specific"

Please revise acknowledgement section "technical assistance and stimulating discussions" may be replaced with "technical assistance and encouragement"

 ---Response：We appreciate these comments and revised as suggested.

Line 337-338: "grafted on specific" was replaced with "grafted onto specific".

Line 341-342: "technical assistance and stimulating discussions" was replaced with "technical assistance and encouragement" in acknowledgement section.

7. PLOS authors have the option to publish the peer review history of their article (what does this mean?). If published, this will include your full peer review and any attached files.

Do you want your identity to be public for this peer review? For information about this choice, including consent withdrawal, please see our Privacy Policy.

Reviewer #1: Yes: Muhammad Azher Nawaz

 ---Response：Thank you. All figure files were uploaded to PACE and no Image Problems were discovered. In revised manuscript, all figures were adjusted according to PACE Adjustments.

---

## [Editor Report · Decision Letter 2]

27 Sep 2019

Metabolomic analysis  of the occurrence of bitter fruits on grafted oriental melon plants

PONE-D-19-19662R2

Dear Dr. Zhao,

We are pleased to inform you that your manuscript has been judged scientifically suitable for publication and will be formally accepted for publication once it complies with all outstanding technical requirements.

With kind regards,

Yuan Huang

Academic Editor

PLOS ONE
---

## [Editor Report · Acceptance letter]

2 Oct 2019

PONE-D-19-19662R2 

Metabolomic analysis of the occurrence of bitter fruits on grafted oriental melon plants 

Dear Dr. Zhao:

I am pleased to inform you that your manuscript has been deemed suitable for publication in PLOS ONE. Congratulations! Your manuscript is now with our production department. 

With kind regards,

on behalf of

Dr. Yuan Huang 

Academic Editor

PLOS ONE